# The Potential of Liquid Biopsy in Detection of Endometrial Cancer Biomarkers: A Pilot Study

**DOI:** 10.3390/ijms24097811

**Published:** 2023-04-25

**Authors:** Dominik Kodada, Michaela Hyblova, Patrik Krumpolec, Nikola Janostiakova, Peter Barath, Marian Grendar, Gabriela Blandova, Oliver Petrovic, Pavol Janega, Vanda Repiska, Gabriel Minarik

**Affiliations:** 1Medirex Group Academy, 94905 Nitra, Slovakia; 2Faculty of Medicine, Comenius University in Bratislava, 84215 Bratislava, Slovakia; 3Jessenius Faculty of Medicine in Martin, Comenius University in Bratislava, 03601 Martin, Slovakia

**Keywords:** endometrial cancer, liquid biopsy, ctDNA, *DNMT3A* mutations

## Abstract

Endometrial cancer belongs to the most common gynecologic cancer types globally, with increasing incidence. There are numerous ways of classifying different cases. The most recent decade has brought advances in molecular classification, which show more accurate prognostic factors and the possibility of personalised adjuvant treatment. In addition, diagnostic approaches lag behind these advances, with methods causing patients discomfort while lacking the reproducibility of tissue sampling for biopsy. Minimally invasive liquid biopsies could therefore represent an alternative screening and diagnostic approach in patients with endometrial cancer. The method could potentially detect molecular changes in this cancer type and identify patients at early stages. In this pilot study, we tested such a detection method based on circulating tumour DNA isolated from the peripheral blood plasma of 21 Slovak endometrial cancer patients. We successfully detected oncomutations in the circulating DNA of every single patient, although the prognostic value of the detected mutations failed to offer certainty. Furthermore, we detected changes associated with clonal hematopoiesis, including *DNMT3A* mutations, which were present in the majority of circulating tumour DNA samples.

## 1. Introduction

Endometrial cancer (EC), which causes malignant changes in the inner layer of the uterine body, is the most common gynaecological cancer in the developed world [1] and the fourth most common malignancy in women worldwide (after breast, lung, and colon cancers) [2,3]. In 2020, the highest cumulative risk of corpus uteri cancer was detected in the regions of North America and Central and Eastern Europe [4]. Among 26 European countries, Slovakia had one of the highest relative incidence rates [5], with an increase in incidence from approximately 1081 new patients diagnosed in 2020 to approximately 1250 cases anticipated in 2040 [4]. In larger developed countries, the increase in incidence is even more rapid [1,2]. Possible causes have been debated, ranging from inadequate research funding [2] to an ageing population (the average age of women diagnosed with EC is 60 years [4]) and an increase in metabolic syndrome and obesity (the predominant risk factor in EC) [6].

In recent decades, many classification systems have been developed that differentiate endometrial tumours according to multiple criteria in an attempt to personalise a wide variety of heterogeneous cases. One of the earlier approaches, which is still in use today, is the World Health Organisation histopathological classification [7]. The most common class is endometrioid adenocarcinoma, which accounts for 80% to 90% of all cases in some populations [5,7]. The second most common type according to said classification is serous adenocarcinoma, which accounts for approximately 10% of all cases. Approximately 1 to 5% of patients with EC have tumours with clear cell histology. The two latter classes are typically associated with a more negative prognosis compared to the more common endometrioid histology. A very small percentage of patients have undifferentiated, dedifferentiated, and mixed EC or carcinosarcoma EC [7]. One of the older classification approaches is based on clinicopathologic types [8]. There are only 2 broad classes primarily differentiated based on various risk factors. The more common type 1 (about 80% of cases) often affects nulliparous women, with later onset of menopause and metabolic syndrome [8,9]. All these risk factors are associated with higher levels of uncompensated oestrogens, which increase endometrial cell proliferation, thus increasing the likelihood of malignant changes at the cellular level [10,11]. Patients with type 2 EC have a worse prognosis, probably due to being diagnosed at an older age. There is no obvious association with higher oestrogen levels, and, given the lower incidence and 5-year survival, they are more difficult to study [8,9,10]. Although this classification was not recommended by the WHO at the clinical or pathological level [7], it proved to be the basis for later molecular classifications as the clinicopathological types have different molecular backgrounds and, thus, different thresholds within the carcinogenic process [12]. One classification approach that has revolutionised prognostication and greatly improved the understanding of changes in EC tumour DNA is the molecular classification proposed in The Cancer Genome Atlas [13]. According to this classification, the POLE category carries mutations in the gene encoding the catalytic subunit of the DNA polymerase Epsilon, which has a role in both replication and repair of the nuclear DNA molecule [14,15] and has a significantly better prognosis than the other categories. The microsatellite instability-hypermutated (MSI-H) category, which is more common than the POLE category, is associated with a moderate prognosis. Their mismatch repair (MMR) activity is deficient either due to DNA methylation or a mutation in one of the genes encoding the MMR protein [13]. Some inherited mutations in the same genes (*MLH1*, *MSH2*, *MSH6*, or *PMS2*) cause Lynch syndrome, the most common genetic syndrome associated with EC. The largest category is copy-number low, with a small number of somatic copy-number changes, and it also has a moderate prognosis. In addition to alterations, it is also associated with mutations in *CTNNB1*. Catenin Beta 1, a product of this gene, is involved in the adhesion protein complex but also binds to the APC protein, which is encoded by the tumour suppressor gene [16]. EC patients with tumours in the copy-number-high category have an even worse prognosis. They carry many somatic copy-number changes and are often associated with mutations in the *TP53* tumour suppressor gene [13].

The most commonly used classification schemes are the International Federation of Gynaecology and Obstetrics (FIGO) classification of grading and staging evaluated after surgical intervention. The current classification system comprises three levels based on the degree of glandular differentiation. The low grades are grades 1 and 2, where tumours show ≤5% and 6% to 50% solid, non-glandular, and non-squamous growth, respectively. The high grade (grade 3) shows more non-glandular growth, which is present in a prognostically high-risk patient group [17,18]. FIGO staging describes the spread of cancer. In stage I, the tumour is confined to the body of the uterus, possibly extending into the myometrium. If the invasion is absent or less than half of the myometrium width is invaded, the stage is determined as stage IA; in other cases, with greater myometrial invasion, the stage is classified as stage IB. In stage II, the tumour spreads to the cervical stroma. Local spread outside the uterus is characteristic of stage III, reaching the uterine serosa, fallopian tubes, and ovaries in stage IIIA, the vagina or parametrium in stage IIIB, and either the pelvic lymph nodes in stage IIIC1 or the paraaortic lymph nodes in stage IIIC2. In the last stage, i.e., stage IV, cancer metastasizes to distant organs, for example, to rectal or bladder tissues in stage IVA or lymph nodes in the groin area and/or distant organs (e.g., the lungs) in stage IVB [19,20].

Approximately three-quarters of cases in postmenopausal women are diagnosed at an early stage, which improves treatment [21]. In both postmenopausal and premenopausal women, vaginal bleeding (although pathognomonic) is the most common clinical manifestation of EC [3,21]. It is recommended that all women with abnormal uterine bleeding over the age of 45 be screened for EC. Additionally, 90% of women diagnosed with EC are symptomatic. In younger women, a history of unsuppressed oestrogen exposure (a major risk factor for EC) is considered [21,22,23]. There are no specific laboratory tests for the evaluation and screening of endometrial cancer, although transvaginal ultrasonography or endometrial biopsy are recommended, depending on the patient and the physician’s preference for availability [3]. Transvaginal ultrasonography is used to measure endometrial thickness. It causes only little discomfort to patients and is indicative mostly of Type 1 tumours [22]. The cut-off size for normal ultrasonography should be either 4 mm or less [24] or 5 mm or less [25]. Alternatively, magnetic resonance imaging can be used to measure endometrial thickness or structural abnormalities. Regardless of the uncertainty in the optimal cut-off measurement, transvaginal ultrasonography is often used as a primary diagnostic approach due to its low cost and high sensitivity [3,25]. Endometrial biopsies are performed to establish a conclusive diagnosis. The two most popular approaches to obtaining endometrial tissue are curettage and the pipelle methods, both of which are relatively invasive and cause some discomfort for patients. However, these methods provide the most accurate diagnosis [22]. The dilation and curettage methods are considered the gold standard, although they carry attendant risks. Endometrial scraping can cause complications including infections, uterine perforation, and bleeding in approximately 60% of cases and must be performed under general anaesthesia [26,27]. Even with hysteroscopy, only 65% of the uterine cavity is sampled [22]. The pipelle method offers a cheaper and less time-consuming alternative with high diagnostic accuracy, but tissue samples are difficult to obtain, with only about one-third of procedures delivering adequate samples [3,28]. The final step in diagnosis is to determine the grade and stage of cancer, which is performed after hysterectomy with bilateral salpingo-oophorectomy, which is the primary treatment for EC [3].

Liquid biopsy has the potential to overcome many of the limitations of the traditional biopsy-based approach in EC diagnosis, mainly due to the low cost and reproducibility of sampling [29]. Fluid collection is minimally invasive, ranging from blood, urine, lavage, cerebrospinal and peritoneal fluids to saliva samples. The analysed samples may include circulating tumour DNA (ctDNA)/cell-free DNA (cfDNA), circulating tumour cells and proteins, and other circulating biomarkers including RNA, vesicles, and platelets [30,31]. Compared to other tumour types (breast, colorectal, cancer, and prostate), research on liquid biopsy in EC is limited and is not used in clinical practice. However, with the advent of molecular classifications, this approach shows the potential to provide diagnostic and prognostic information that could lead to screening, monitoring, and proper stratification of patients with EC [30,31,32]. The latest studies reveal that EC biomarkers are detectable, especially at the later stages because of the increased quantity of ctDNA/cfDNA found in blood [32,33]. 94% of patients (out of 48) in the study by Bolivar et al. have at least one EC oncomutation, mostly in the *CTNNB1*, *KRAS*, *PTEN*, and *PIK3CA* genes. However, only in one out of three of these patients was the same mutation also present in the tumour [32]. Another, larger study (193 patients) by Danziger et al. comprised 94% of EC patients with detectable ctDNA [34]. No such study has ever been tested in Slovak EC patients.

Although its results are being incorporated into clinical cancer detection, the use of liquid biopsy still comes with many limitations. It is unable to distinguish mutations in tumour tissues from somatic mosaicism caused by the abundance of free cellular DNA derived from haematopoietic cells, which constitutes the biological background noise of the blood-based liquid biopsy. In elderly patients, age-related clonal haematopoiesis (ARCH) often develops on the basis of somatic mutations and clonal expansion of mutant stem cells [35]. Approximately 10% of people over the age of 70 years carry mutations associated with ARCH, with a higher frequency found in men compared to women [36]. The most commonly mutated ARCH-related genes are *DNMT3A*, *TET2*, *JAK2*, *GNAS*, and *TP53*, although all of them can be significant as cancer-related genes as well. It is unclear what the clinical significance is when these mutations are found in the circulating blood of cancer patients [35].

## 2. Results

### 2.1. Tumour Mutational Burden and Microsatellite Instability

TruSight Oncology’s 500 libraries after NGS sequencing allow quantification of total tumour mutational burden (TMB) as the number of mutations per Mb. High overall TMB was present in only one ctDNA-FFPE-DNA pair, reaching 32.1 and 26.3 mutations per Mb, respectively. High TMB was present at the FFPE-DNA level in 3 other samples (28.2, 41.5, and 89.6 mutations per Mb) and only at the ctDNA level in 2 other samples (37.2 and 50.2 mutations per Mb) (Table 1). In addition to the detection of polymorphisms and TMB, microsatellite instability (MSI) was also evaluated. MSI results from impaired DNA error repair that may be a manifestation of Lynch syndrome but may also make tumours more sensitive to specific therapy with cell cycle checkpoint inhibitors. With a cut-off of 20% unstable MSI sites, we identified two patients with high-grade MSI, in both cases only at the FFPE-DNA level (Table 1).

### 2.2. DNA Mutation Detection

In each patient, we successfully identified mutations with potential clinical relevance (IVA level 2C) for EC at least at the ctDNA level, as we did not detect any potentially clinically relevant oncomutations in the three FFPE-DNA samples. The average number of mutations for each stage and grade is shown in Table 2. The majority of identified mutations were single nucleotide variants and small-scale deletions and insertions that have a loss of function effect. Overall, we identified mutations in more than 70 genes and at least one common potentially clinically relevant mutation in two-thirds of patients (14/21). In 2 out of the 5 high-grade/high-risk samples (G3), no potential clinically significant mutations were found at the FFPE-DNA level, only at the ctDNA level. Identical mutations in this high-risk category were present in only 2 ctDNA-FFPE-DNA sample pairs.

Of the 2C-tier mutations identified from the 21 FFPE-DNAs, mutations in the *PTEN* (48%), *ARID1A* (29%), *PIK3CA* (29%), *PIK3R1* (24%), *CTCF* (24%), *BRCA2* (24%), *ZFHX3* (19%), *CTNNB1* (14%), *FGFR2* (14%), *SOX17* (14%), and *CHEK2* (14%) genes were the most common (Table 3). At the ctDNA level, mutations in the *DNMT3A* (52%) and *TET2* (14%) genes were unusually frequent, which we did not identify at the FFPE-DNA level in any of the cases. However, they occurred in four of the five samples with high-grade tumours. The next most frequently mutated genes at the ctDNA level included *CHEK2* (24%), *PTEN* (19%), *ZFHX3* (19%), *PIK3CA* (14%), and *TP53* (14%) (Table 3).

### 2.3. Mutations in the DNMT3A Gene

Altogether, we identified 144 protein variants for the *DNMT3A* gene from the ctDNA samples, 33 of which were assigned as pathogenic. Almost 70% of identified pathogenic variants were localised in functional domains: 9% in the PWWP domain, 18% in the ADD domain, and 42% in the SAM-dependent MTase C5-type domain (Figure 1).

## 3. Discussion

A benefit of the TSO500 sequencing approach is its ability to quantify TMB and % MSI sites. High TMB, with a cut-off point of >15.5 mutations per Mb, has been reported to be relevant as it increases the efficacy of checkpoint inhibitor immunotherapy [53], and these patients thus have better overall survival [54,55]. A high MSI is mostly associated with the MMR-deficient molecular classification category. Its presence also improves the response to antibody-based cancer therapy [55]. In our dataset, a high number of MSI sites could not be detected at the ctDNA level, which may be due to background noise caused by the heteromolecular content of cfDNA in blood plasma. Of the two tumours with MSI-H values, one did not carry any clinically relevant MMR gene mutations. A possible explanation is the frequently occurring hypermethylation of the *MLH1* gene promoter in MMR-deficient tumours. Detection of methylation is a factor limiting our approach and has the potential to improve future screening approaches, as ctDNA methylation has been shown to be sensitive in detecting biomarkers of other malignancies [56].

The second patient (#19) with a high percentage of MSI sites carries several prognostically conflicting DNA mutations. With a *POLE* mutation that is prognostically positive, 2 biallelic *MSH6* mutations that have moderately favourable outcomes, and *TP53* mutations with an unfavourable prognosis according to PORTEC studies [57], the case is considered a multiple classifier. Triple multiple classifiers are very rare, even in studies comprising severalfold larger samples (less than 0.5% of patients), so survival prognosis data is limited [58], but other *POLE* mutations containing multiple classifiers show a favourable prognosis [18,58]. In addition, the presence of two biallelic *MSH6* mutations may suggest the possibility that they may not all be somatic, but one (or both) may have been inherited, making patient 19 a possible case of Lynch syndrome. Mutations in *MSH6* cause approximately 10% of Lynch syndrome cases [59], and as 3% of EC cases are associated with Lynch syndrome [60], this would be consistent with the literature and the size of our cohort. Moreover, *MSH6* mutations are also linked with a presentation at a higher age in both EC and gastrointestinal cancer [61,62], and this patient was 73 years old during the surgery. Unfortunately, we did not sample enough peripheral blood from the patient to prove a case of Lynch syndrome.

Although our pilot study is relatively small, our cohort turns out to be representative across several features. Endometrioid histology is most common in EC, and the mean age of our cohort (64.77 years) is comparable to the overall mean age at diagnosis (61 years) [63]. To our knowledge, there is no widely applied screening approach among Slovak gynaecologists, and therefore the mean age of patients was expected to be higher. The large proportion has early-stage and low-grade tumours (Table 1), which is consistent with the general female population as well as with women of Caucasian descent [64]. Overall, we identified oncomutations in 73 genes. As expected, we identified more oncogenic EC-related mutations in tumour DNA samples, some of which were prognostically relevant. We found *POLE* mutations in two patients, MSI was estimated to be high in two cases, one additional case had a mutation in the *MLH1* MMR gene, and *TP53* mutations were found in three tumours, which is the same number as for tumours with *CTNNB1* mutations. Most patients would therefore be considered to have a moderate prognosis (including patient no. 19 with the triple multi-classifier); two patients would be in the unfavourable *TP53* category and one in the favourable *POLE* mutation group. The most common tumour mutations were in the *PTEN* gene, which is similar to other studies in which 40–50% of all EC cases carry such mutations, especially those with endometrioid histology. The *PTEN* mutations are associated with an early stage, and the prognosis for these patients is mostly favourable. Notably, in both tumours with *POLE* mutations, *PTEN* mutations were also present [65]. Both *POLE* and *ARID1A* genes are tumour suppressors, but the latter is not associated with any specific histology [66]. The *PIK3CA* and *PIK3R1* genes encode subunits of the PI3K kinase, which is part of a signalling pathway that regulates cell survival and apoptosis. *PIK3CA* mutations are particularly common in tumours with low-grade and early-stage endometrioid histology [67]. We found these mutations in six tumour tissues, one of which is a high-grade IIIA case.

Despite the representativeness of the cohort across several indicators and the tumour’s molecular background, the ctDNA sequencing from peripheral blood revealed a slightly different molecular profile. In the above-mentioned similar study by Bolivar et al., several mutations in the same genes were identified (*CTNNB1*, *KRAS*, *PTEN*, or *PIK3CA*) [32]. In a different study that was monitoring the changes in blood ctDNA in response to post-surgical treatment of gynecologic malignancies, *PTEN*, *PIK3CA*, *TP53*, and *KRAS* genes were detected, although these patients were mostly treated for high-grade EC and were not limited to endometrioid histology [68]. It suggests that the method may be more sensitive to mutations in these genes. In two-thirds of all patients, we were able to detect at least one common oncomutation between pairs of ctDNAs and FFPE-DNA samples, which is relatively high compared to Bolivar et al. [32], who detected it in one-third of their patients. Further, there was at least one EC oncomutation in every ctDNA sample of our patients, regardless of tumour stage or grade (Table 2). Despite the limited size of the dataset, these results seem to suggest that blood ctDNA sequencing offers some potential for EC screening. The detection of prognostically important mutations of EC genes in blood ctDNA failed to show potential in our study, as there were no *POLE* mutations, no high MSI, and none of the *TP53* mutations detected matched the mutations in the FFPE-DNA of the patients. Despite its undeniable advantages, including minimal invasiveness and reproducibility with greater objective representativeness than curettage-derived excisions, the potential diagnosis of EC using blood ctDNA has the disadvantage of detecting the molecular background noise of the whole body. Several of the cancer-related genes we identified, including *PIK3CA*, *KRAS*, *GNAS*, and *PPM1D*, could also have haematological mutations. In addition, mutations in *DNMT3A*, *TET2*, *TP53*, and *SF3B1* are among the most common genes involved in clonal hematopoiesis in healthy individuals [35]. Therefore, other sources of ctDNA could represent alternatives free of molecular background noise, such as urine or uterine lavage. Uterine lavage is a preferable option because of the direct contact of the fluid with the endometrial tumour, although its disadvantages include greater invasiveness and patient discomfort.

Given the higher age of the cohort, it was not surprising that we found some mutations in ARCH genes. They were usually found sporadically (in 1–2 cases), except for *PIK3CA*; however, these mutations were also common in tumour tissue and *TP53*, which are discussed above. The presence of *DNMT3A* mutations in the ctDNA of 11 patients and the absence of any of them in FFPE-DNA represented a notable finding. Mutations in *DNMT3A* are most common in ARCH, which occurs in approximately 5% of the population over 60 years of age [69]. Of the 18 patients who were older than 60 years, we found them in 10 cases (55.56%). This more than 11-fold increase in incidence compared to the general population may be explained by the late age of the patients and the relatively small size of the cohort; however, we observed an association of their presence with high-grade tumours (they were present in 3 of the 5 G3 patients, while 1 of the remaining 2 patients carried the *TET2* mutation in the ctDNA). Therefore, we propose that the most frequently occurring ARCH mutations (in *DNMT3A* and *TET2*) might represent a negative prognostic marker for EC. However, this hypothesis requires more extensive research.

Overall, mutations in *DNMT3A* and *TET2* can disrupt normal DNA methylation patterns, which may lead to dysregulation of gene expression. Dnmt3a is a de novo methyltransferase [70], and Tet2 oxidises 5-methylcytosine to 5-hydroxymethylcytosine, which is also an epigenetic DNA modification process [52]. However, it is important to note that the exact mechanisms by which these mutations contribute to cancer development and progression are not yet fully understood. They may vary depending on the specific type of cancer and the context in which the mutations occur. Mutations in *DNMT3A* have been found in several types of cancer, including acute myeloid leukaemia, myelodysplastic syndromes, and other haematological malignancies. These mutations can lead to abnormal DNA methylation patterns and contribute to the development and progression of cancer [71]. Mutations in *TET2* have been associated with EC development as well as a negative prognostic marker [52]. Additionally, another study found that mutations in *TET2* were associated with the activation of the PI3K-Akt-mTOR signalling pathway, which is known to be involved in cell proliferation and survival [72].

The wild-type Dnmt3a protein contains three highly conserved main domains: the PWWP domain, required for association with pericentric heterochromatin; the ADD domain, which interacts with the H3 histone 1–19 tail; and the large C-terminal SAM-dependent MTase C5-type domain, required for methyltransferase activity of the protein [73]. The majority (69.7%) of identified pathogenic variants in our study have missense mutations in these functional domains (Figure 1). The presence of *DNMT3A* mutations in blood ctDNA was previously observed among EC patients in a recent study by Danziger et al. [34]. Unfortunately, the authors do not specify the number of samples carrying each mutation; they mention only a significantly larger number of ctDNA samples than tumour DNA samples. Moreover, they do not provide any prognostic information about those patients separately. Interestingly, *DNMT3A* gene hypermethylation was found to be an important event in EC carcinogenesis [70].

## 4. Materials and Methods

The study included 21 Slovak patients with EC who underwent an abdominal hysterectomy. It was conducted in accordance with the Declaration of Helsinki, and the protocol was approved by the Ethics Committee of the Bratislava Self-Governing Region (06196/2020/HF). Written informed consent was obtained from all participants.

The age range of the patients was 34–84 years, with a mean age of 64.77 years and a median age of 63.5 years. All tumours were confirmed by endometrioid histology. The detailed characteristics of the study population are shown in Table 1.

Furthermore, we analysed peripheral blood samples collected before surgery and tumour tissues that were in formalin-fixed paraffin-embedded (FFPE) blocks. Blood samples were centrifuged (15 min; 2200× *g*); plasma was post-extracted and stored at −20 °C until further processing. Circulating tumour DNA was extracted from 2 mL of blood plasma samples using the QiaSymphony SP automated system (QIAGEN, Redwood City, CA, USA) with magnetic bead extraction (QIASymphony DSP Circulating DNA kit). The FFPE tumour tissue blocks were cut into 10 µm sections, which were then used for DNA isolation using the Ionic FFPE to Pure DNA kit and for RNA isolation using the Ionic FFPE to Pure RNA kit. The TruSight Oncology 500 and TruSight Oncology 500 ctDNA protocols (Illumina) were used to prepare genomic sequencing libraries. This method enables 24 samples to be analysed simultaneously in a single run with sufficient coverage to identify mutations at the FFPE and ctDNA levels and allows the identification of small and CNV variants in addition to TMB and MSI within a single analysis. Samples were analysed randomly in the order included in the study to fulfil the capacity of the sequencing chip (21 pairs complied with the qualitative and quantitative metrics for evaluation). No sample was prioritised based on tumour type, stage, grade, or the patient’s age. The libraries were then sequenced on the NovaSeq 6000 platform (Illumina), and primary analysis, tumour mutational load, microsatellite instability status, and mapping were performed according to the TSO500 alignment and variant calling described by [53] and annotated using Ingenuity Variant Analysis (IVA) software (QIAGEN, Redwood City, CA, USA, Software Build: 9.1.1.20230406).

## 5. Conclusions

In our pilot study, we detected at least one oncomutation in ctDNA from peripheral blood, making liquid biopsy a potential approach to diagnosing EC. To our knowledge, no similar study has been performed on the Slovak population. Although the identified cancer-related mutations have very limited prognostic value, the method may supplement an EC screening approach. An improvement in the performance of liquid biopsy could be achieved by using alternative sources of ctDNA (uterine lavage). Moreover, in accordance with our results, we hypothesise that ARCH-related mutations (in the *DNMT3A* and *TET2* genes) could be associated with a worse prognosis for endometrial cancer patients. Especially the *DNMT3A* mutations were detected significantly more often in the ctDNA of our patients. Further research is required to shed more light on the molecular mechanisms of the ARCH and EC interplay.

## Figures and Tables

**Figure 1 ijms-24-07811-f001:**
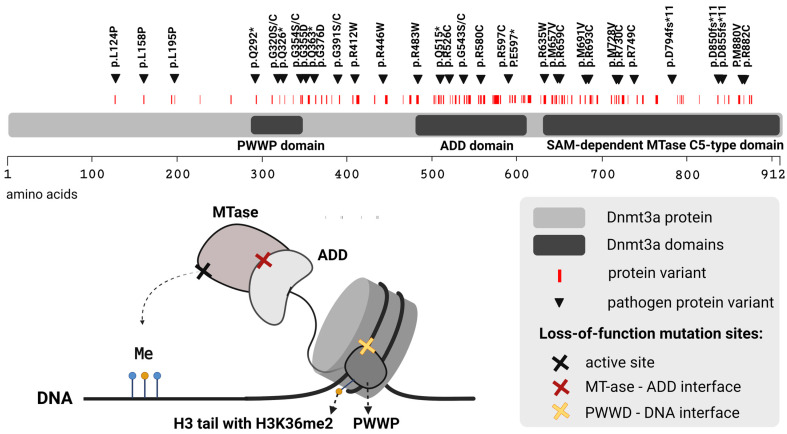
Dnmt3a protein variant location. The figure shows the location of all identified Dnmt3a protein variants (labelled by **I**) as well as pathogenic Dnmt3a protein variants (labelled by ▼). The loss-of-function mutation sites are also highlighted (labelled by **✕**) at Dnmt3a domains related to identified pathogenic protein variants. The figure was created with BioRender.com (22 March 2023).

**Table 1 ijms-24-07811-t001:** Cohort characteristics, including TMB and MSI values in FFPE-DNA and ctDNA libraries (the high total TMB and high % of MSI highlighted in blue), of our study patients and specifics of their age and tumour classifications.

Patient No.	Age at Surgery	Tumour Histology	FIGO Grade and Stage	Total TMB in FFPE-DNA Libraries	Total TMB of ctDNA Libraries	% of MSI Sites in FFPE-DNA Libraries	% of MSI Sites in ctDNA Libraries
01	64	endometrioid	G3 I A	0.80	2.50	0	0
02	63	endometrioid	G2 I A	3.90	9.80	3.39	0
03	70	endometrioid	G3 III A	0.80	5.10	1.68	0.05
04	75	endometrioid	G2 I A	1.60	4.20	0	0.09
05	58	endometrioid	G2 I A	1.60	3.10	5.22	0
06	60	endometrioid	G2 I A	1.60	4.20	0.85	0.5
07	68	endometrioid	G2 I A	1.60	0	1.64	0
08	76	endometrioid	G2 I B	16.40	3.30	13.54	0.05
09	84	endometrioid	G2 I B	32.10	26.30	25.20	0.46
10	61	endometrioid	G2 II	9.40	6.00	2.46	0.14
11	60	endometrioid	G1 I A	0	5.20	1.69	0
12	60	endometrioid	G1 I A	6.30	0	3.45	0
13	34	endometrioid	G1 I A	1.60	2.50	1.64	0
14	63	endometrioid	G2 I B	28.20	3.30	0.84	0.09
15	57	endometrioid	G2 I A	0	1.70	0	0.05
16	63	endometrioid	G3 I B	7.10	0.80	2.32	0
17	84	endometrioid	G2 I B	8.60	11.60	0.83	0
18	64	endometrioid	G3 I B	0	37.20	1.68	0.23
19	73	endometrioid	G1 I A	41.50	2.40	42.28	0.14
20	68	endometrioid	G3 III A	0	50.20	0.80	0.81
21	64	endometrioid	G2 I B	89.60	9.10	0.04	0.04

**Table 2 ijms-24-07811-t002:** The average number of clinically significant mutations found at each stage (**A**) and each grade (**B**).

(**A**)	**Stages**	**Mutations FFPE-DNA**	**Mutations ctDNA**	(**B**)	**Grade**	**Mutations FFPE-DNA**	**Mutations ctDNA**
	I A (11)	4.09	2.73		G1 (4)	5.75	1.75
	I B (7)	6.86	5.43		G2 (12)	5.58	4.25
	II (1)	5.00	2.00		low (16)	5.63	3.63
	III A (2)	1.00	9.00		G3 (5)	2.00	6.00

**Table 3 ijms-24-07811-t003:** List of most frequently mutated genes from our study with their product functions.

Gene Name	Frequency in FFPE-DNA	Frequency in ctDNA	Gene Product Function	Reference
*PTEN*	48%	19%	protein/lipid phosphatase, tumour suppressor	[37]
*ARID1A*	29%	10%	chromatin remodelling, tumour suppressor	[38]
*PIK3CA*	29%	14%	kinase subunit, the most frequently mutated oncogene in human cancers	[39]
*PIK3R1*	24%	14%	kinase subunit, tumour suppressor	[40]
*CTCF*	24%	10%	genome stability, tumour suppressor	[41]
*BRCA2*	24%	10%	genome stability, tumour suppressor	[42]
*ZFHX3*	19%	19%	transcription factor, tumour suppressor	[43]
*TP53*	14%	14%	cellular response to stresses, maintenance of genomic integrity, tumour suppressor	[44]
*FGFR2*	14%	5%	signalling, tumour suppressor	[45]
*CTNNB1*	14%	5%	cell growth and adhesion, signalling	[46]
*SOX17*	14%	0%	β-catenin inhibitor, tumour suppressor	[47]
*CHEK2*	14%	24%	protein kinase, tumour suppressor	[48]
*FAT*	10%	0%	cell adhesion and extracellular matrix architecture regulation, tumour suppressor	[49]
*KRAS*	5%	5%	signalling, oncogene	[50]
*DNMT3A*	0%	52%	DNA methylation, ARCH biomarker	[51]
*TET2*	0%	14%	methylation regulation, tumour suppressor, ARCH biomarker	[52]

## Data Availability

Data from genomic sequencing are available on request.

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
