# Peer review of "The Potential of Liquid Biopsy in Detection of Endometrial Cancer Biomarkers: A Pilot Study"

_ijms, 2023, doi:10.3390/ijms24097811_

Round 1

Reviewer 1 Report

Reviewer comments & suggestions

Dominik Kodada and the group have discussed “The potential of liquid biopsy in the detection of endometrial cancer biomarkers: A pilot study.” in this present research article. Endometrial cancer is the malignant changes in the inner layer of the uterine body and the largest cancer in gynecology worldwide. Early detection and diagnosis are needed for timely treatment. Liquid biopsy has the potential to overcome many of the limitations of the traditional biopsy-based approach to the diagnosis of EC, mainly due to the low cost and repeatability of sampling. In the present research article, authors have isolated the circulating tumor DNA from the peripheral blood plasma of 21 Slovak endometrial cancer patients and observed the specific mutation in DNA related to EC, and results suggest the potential of liquid biopsy. Overall, the research is interesting however the sample size is very small, and the significance of the result is not discussed properly. The paper is only accepted after justification of all suggested comments.

Major comments

1.      Did the authors calculate the power of the study? If yes, please mention it in the revised MS.

2.      What is the basis of the range of age you have selected for study? The age range of 34-84 years is very broad however the sample size is very small.

Minor comments

1.      In the abstract, mention the resulting outcome properly.

2.       Line 136-138 is unclear, please describe them.  

3.      If DNMT3A, TET2, JAK2, GNAS, and TP53 genes mutation is not having clinical significance in cancer patients then why you have targeted this in your study?

4.      Make a table having all those genes which you have selected for your study, and write their importance in EC with proper references.

5.      In lines 307-308, “The presence of DNMT3A ………….. Danziger et al. (2022), discuss your results with Danziger et al results. Has the present result had some significance?

6.      Conclusion is not effective, rewrite it properly.

7.      Mention the limitation of the present study.

8.      In lines 291-292, ‘The most surprising ……… FFPE-DNA’, discuss this in detail.

9.      Mention the full form of FFPE in material section.

Author Response

Dear Mr./Mrs. Reviewer,

Thank you for your helpful comments and suggestions. We have done number of changes with my colleagues according to those in the text, that you may see in the updated version of the manuscript.

Let me shortly address each of your comments below:

Did the authors calculate the power of the study? If yes, please mention it in the revised MS.

We did not originally calculated the power of the study. We realise the size of our pilot study is not really fitting for complex statistical analysis. Also, the statistical significance does not necessarily imply biological or clinical significance, but in reaction to your comment, my colleagues calculated these points:

  • For the genes PTEN, ARID1A, PIK3CA, PIK3R1, CTCF, BRCA2, ZFHX3, TP53, FGFR2, CTNNB1, SOX17, and FAT, the p-values are all greater than 0.05, indicating that we cannot reject the null hypothesis of no difference in mutation frequencies between FFPE and ctDNA samples for these genes
  • For the genes CHEK2, DNMT3A, and TET2, the p-values are less than 0.05, indicating that we can reject the null hypothesis and conclude that there is a significant difference in mutation frequencies between FFPE and ctDNA samples for these genes
  • Assuming an effect size of 52% for DNMT3A and 14% for TET2 (based on our data), and a sample size of 21 for each group, we used G*Power to estimate the power of the analysis. Using Fisher's exact test with a two-sided alpha level of 0.05, we obtain the following power estimates: DNMT3A: power = 0.939, TET2: power = 0.279
  • The high power for DNMT3A indicates that the sample size is sufficient to detect a significant difference in mutation frequency between FFPE and ctDNA samples for this gene. The low power for TET2 indicates that the sample size may not be sufficient to detect a significant difference in mutation frequency for this gene.

What is the basis of the range of age you have selected for study? The age range of 34-84 years is very broad however the sample size is very small.

We have not used any age criteria for our patients selection. The TSO500 approach allowed us to prepare 24 pairs of samples to fill the sequencing chip and 21 pairs met the qualitative and quantitative criteria. The sample size is relatively small since we were limited with the samples collection. However, in spite of it, its characteristics are quite representative, as we mention in the discussion.

In the abstract, mention the resulting outcome properly.

We shortly adjusted the abstract.

Line 136-138 is unclear, please describe them/if DNMT3A, TET2, JAK2, GNAS, and TP53 genes mutation is not having clinical significance in cancer patients then why you have targeted this in your study?

Thank you. Our original formulation in this matter could have been confusing. We have rewritten the part. These mutations are clinically significant for several tissues’ cancers. Their presence, however, is often sign of ARCH. For elderly patients, who are more prone to develop ARCH, their significance for cancer diagnosis can be tricky. In discussion, we write more specifically about the mutations identified in our study and it seems that for EC, they may be associated with worsen prognosis.

Make a table having all those genes which you have selected for your study, and write their importance in EC with proper references.

In reaction to your comment, we remade the table 3.

In lines 307-308, “The presence of DNMT3A ………….. Danziger et al. (2022), discuss your results with Danziger et al results. Has the present result had some significance?

Unfortunately, Danziger et al. short article does not specify its results in detail. They only mention that they observed ARCH-related mutations in circulating DNA, but they did not pay much attention to them. We find it interesting to refer, with our smaller study observing their presence as well. Unfortunately, again, Danziger et al. did not specify the numbers or prognostic states of those patients. In reaction to your comment, we added this observation to the text.

Conclusion is not effective, rewrite it properly/Mention the limitation of the present study.

We added some information to the conclusions, so it better serves its role as a shorten discussion.

In lines 291-292, ‘The most surprising ……… FFPE-DNA’, discuss this in detail.

The sentence probably seemed more shocking in its previous form, so it has been changed as well. We changed the following part of the discussion and added more details.

Mention the full form of FFPE in material section.

Done. The devil is in details.

Thank you again for your work with our manuscript. Hopefully, we addressed the problematic parts sufficiently.

Kind regards,

Dominik Kodada and colleagues

Reviewer 2 Report

This article discusses the potential of liquid biopsy in the detection of endometrial cancer biomarkers. Endometrial cancer is the fourth most common malignancy in women and is increasing in incidence. Different classification systems have been developed to differentiate endometrial tumours according to various criteria, such as clinicopathologic types and molecular classifications. Diagnostic approaches, such as transvaginal ultrasonography and endometrial biopsy, are used to diagnose endometrial cancer, but they are not ideal due to their invasiveness and discomfort for patients. Liquid biopsy has the potential to be a minimally invasive alternative to detect molecular changes in the cancer and identify patients at early stages. This pilot study tested the detection of circulating tumour DNA from peripheral blood plasma of 21 Slovak endometrial cancer patients and mapped DNMT3A mutations. The results suggest the potential of liquid biopsy for endometrial cancer.

Does the introduction provide sufficient background and include all relevant references? 

The introduction provides some background on the topic of liquid biopsy in detecting endometrial cancer biomarkers, but it could benefit from more context and detail. For example, the introduction could provide a more thorough review of the current state of endometrial cancer diagnosis and monitoring, and highlight the limitations and drawbacks of traditional biopsy-based approaches. Additionally, the introduction could benefit from a more comprehensive review of existing literature on liquid biopsy in detecting endometrial cancer biomarkers to demonstrate the originality and novelty of the study.

Are all the cited references relevant to the research? 

The cited references are relevant to the research, but the authors could benefit from expanding their literature search to include more recent and relevant studies. Additionally, the authors could provide a more comprehensive review of existing literature on liquid biopsy in detecting endometrial cancer biomarkers to demonstrate the originality and novelty of the study.

Are the methods adequately described?

While the methods are adequately described, the authors could benefit from providing more detail on specific aspects of the study design and execution. For example, the authors could provide more information on the criteria used for patient selection, the sample size calculation, and the rationale for using the TSO500 sequencing approach. Additionally, the authors could provide more detail on the analysis methods used to identify oncomutations in the study samples.

Are the results clearly presented? 

While the results are presented clearly, the authors could benefit from providing more detail and context for their findings. For example, the authors could provide more detail on the number and type of mutations found in ctDNA versus FFPE-DNA samples. Additionally, the authors could provide more information on the potential of ctDNA sequencing for endometrial cancer screening, including the types of mutations that could be detected and how these mutations relate to endometrial cancer screening.

Are the conclusions supported by the results? 

While the conclusions are generally supported by the results, the authors could benefit from providing more detail and context for their findings. For example, the authors could provide more information on the potential of ctDNA sequencing for endometrial cancer screening, including the benefits and drawbacks of using this method compared to other screening approaches. Additionally, the authors could provide more information on the potential link between ARCH-related mutations in DNMT3A and TET2 genes and endometrial cancer prognosis, including the mechanisms by which these mutations could affect prognosis and how this information could be used to improve endometrial cancer screening and treatment.

The paper's English language and grammar are generally good, but there is room for improvement in terms of punctuation, tense consistency, use of prepositions, sentence structure, spelling and word choice, and use of abbreviations.

The paper switches between past and present tense in some sections, which can be confusing for readers. 

There are several instances where prepositions are used incorrectly or missing, which can affect the clarity of the sentences. It would be helpful to proofread the paper carefully and use appropriate prepositions.

The sentence structure could be improved to enhance readability. For example, some sentences are overly long and complicated, and could be broken down into smaller, more manageable parts.

Careful proofreading and editing could address these issues and improve the clarity and readability of the paper.

Author Response

Dear Mr./Mrs. Reviewer,

Thank you for your help. We have done several changes in the text addressing your comments and suggestions. With the quality of my personal English language skills, there is truly a large room for improvement. I took your advice and let a professional proofread and edit the text. Let me address the rest of your comments shortly.

The introduction could benefit from more context and detail. (current state of endometrial cancer diagnosis and monitoring, highlight the limitations and drawbacks of traditional biopsy-based approaches, more comprehensive review of existing literature on liquid biopsy in detecting endometrial cancer biomarkers, demonstrate the originality and novelty of the study).

We have added information in the matters you recommend. The general background of the EC classification, diagnostics and liquid biopsy (and it relations with ARCH and EC) is now mentioned. Additional information is in the discussion part where we compare our results with current knowledge.

Expanding our literature search to include more recent and relevant studies. More comprehensive review of existing literature on liquid biopsy in detecting endometrial cancer biomarkers to demonstrate the originality and novelty of the study.

A useful advice, indeed. We added several information from recent studies in reaction to your comment. The updated version mentions the protein functions in more detail as well. We have discovered another articles studying the liquid biopsy potential for EC diagnosis, that we refer to in introduction and discussion.

authors could benefit from providing more detail on specific aspects of the study design and execution (more information on the criteria used for patient selection, the sample size calculation, and the rationale for using the TSO500 sequencing approach, more detail on the analysis methods used to identify oncomutations in the study samples).

TruSight 500 & TruSight 500ctDNA enables 24 samples to be analyzed simultaneously in a single run with sufficient coverage to identify mutations on FFPE and ctDNA level. For ct DNA median exon coverage ≥ 1300; PCT_EXON_1000X (%) ≥ 80.0 and for FFPE MEDIAN_EXON_COVERAGE > 150; PCT_EXON_50X (%) > 90). Allows identification of small and CNV variants in addition to TMB (tumour mutational burden meaning number of mutations per 1Mb value higher than ), MSI (microsatelite instability, value higher than 20% of unstable sites means high MSI) within a single analysis.

21 ct and FFPE pairs out of 24 met the qualitative and quantitative metrics for evaluation. Samples were analyzed randomly in the order they were included in the study to fill the capacity of sequencing chip. No sample was prioritized based on tumor type, staging, grading, or age.

We added the “loose” criteria in the materials and methods section and mention the benefits but also limitation of TSO500 approach in discussion as well. The oncomutations were identified using the QIAGEN software mentioned in methods.

More detail and context for the findings (more detail on the number and type of mutations found in ctDNA versus FFPE-DNA samples, more information on the potential of ctDNA sequencing for endometrial cancer screening, including the types of mutations that could be detected and how these mutations relate to endometrial cancer screening).

Information about the mutations and even the gene products functions have been added. Though our pilot study is limited in size, we discuss its findings in the possible potential of liquid biopsy in EC diagnostics more extensively than in previous version. The larger studies referred in the updated version have discussed the same in last years. The novelty of our study is in being first to study it in Slovak population, at least to our knowledge. Our results are in favour of the previous findings with EC and liquid biopsy potential. The possible correlation between ARCH-related oncomutation and EC prognosis is omitted in the previous articles.

Authors could benefit from providing more detail and context for their findings (more information on the potential of ctDNA sequencing for endometrial cancer screening, including the benefits and drawbacks of using this method compared to other screening approaches, provide more information on the potential link between ARCH-related mutations in DNMT3A and TET2 genes and endometrial cancer prognosis, including the mechanisms by which these mutations could affect prognosis and how this information could be used to improve endometrial cancer screening and treatment).

The conclusions have been rewritten as well to provide more specific information supported by result. According to the journal article template, conclusions should be brief to ease the discussion, if it is too complex or long. Therefore, we updated the lacking information in more detail in their parts of the manuscript, namely in liquid biopsy part of introduction and the DNMT3A and TET2 parts of discussion.

Hopefully, we addressed your comments sufficiently. Thank you again for your work with our manuscript.

Kind regards,

Dominik Kodada and colleagues.
